# Multidimensional Bayesian Utility Maximization: Tight Approximations to Welfare

Kira Goldner [*]

Taylor Lundy [†]

## Abstract

We initiate the study of multidimensional Bayesian utility maximization, focusing on the unit-demand setting where values are i.i.d. across both items and buyers. The seminal result of Hartline and Roughgarden '08 studies simple, information-robust mechanisms that maximize utility for $n$ i.i.d. agents and $m$ identical items via an approximation to social welfare as an upper bound, and they prove the gap between optimal utility and social welfare is $\Theta(1 + \log n/m)$ in this setting. We extend these results to the multidimensional setting. To do so, we develop simple, prior-independent, approximately-optimal mechanisms, targeting the simplest benchmark of optimal welfare. We give a $(1-1/e)$-approximation when there are more items than buyers, and a $\Theta(\log n/m)$-approximation when there are more buyers than items, and we prove that this bound is tight in both $n$ and $m$ by reducing the i.i.d. unit-demand setting to the identical items setting. Finally, we include an extensive discussion section on why Bayesian utility maximization is a promising research direction. In particular, we characterize complexities in this setting that defy our intuition from the welfare and revenue literature, and motivate why coming up with a better benchmark than welfare is a hard problem itself.

## 1 Introduction

Utility maximization—also sometimes called consumer surplus, residual surplus, money burning, or a setting with ordeals—is best motivated by a social service provider who wishes to allocate goods whose demand far exceeds their supply (or budget), such as medications and vaccines. In such cases, it would be inequitable to gate-keep these goods by charging a price. Instead, acquiring these items is often made laborious by requiring consumers to wait in long lines, fill out forms, travel physical distances, or even try other medications first. These *ordeals* act like payments by ensuring that only those who have high value for the good are actually able to obtain it. As with social welfare maximization, the social service provider has the option to use (non-monetary) payments as a tool in order to allocate items to those with higher need; in contrast, in this setting, doing so comes at a cost to the objective function, as neither consumer nor seller benefit from these ordeals. Crucially, this yields a trade-off in the objective function between allocation and payment—is it more important to give the goods to the right people, potentially using expensive payments that decrease utility to do so, or to minimize payments, allowing the allocation to be more random? Despite sitting between social welfare maximization and payment minimization, intuition from both regimes can fail, requiring clever solutions somewhere in between.

[*]Boston University; `goldner@bu.edu`. Supported by NSF Award CNS-2228610, NSF CAREER Award CCF-2441071, and a Shibulal Family Career Development Professorship.

[†]University of British Columbia; `tlundy@cs.ubc.ca`. Supported by an NSERC Discovery Grant, and a CIFAR Canada AI Research Chair (Alberta Machine Intelligence Institute).

A preliminary version of this work appeared on arXiv under the title "Simple Mechanisms for Utility Maximization: Approximating Welfare in the I.I.D. Unit-Demand Setting."

In the single-dimensional setting, utility-optimal mechanism design is solved by a closed-form theory [Hartline and Roughgarden, 2008]. In this seminal paper, Hartline and Roughgarden study the setting where there are $m$ identical items and $n$ i.i.d. agents. In addition to other contributions, they design a prior-free mechanism that achieves an $O(1 + \log \frac{n}{m})$-approximation of utility to social welfare, which they prove is tight.

In the multidimensional setting, the problem becomes unwieldy, much like revenue maximization. Nothing is known here on the structure of optimal mechanisms. And, unlike revenue, the single-bidder multidimensional problem does not give us a foothold, as the tension between efficient allocation and using payments comes only from limited supply between agents. In this paper, we initiate the study of multidimensional utility maximization from a Bayesian mechanism design perspective. Given the complexities of this setting (discussed more in Section 4 and Appendix D), our main approach is thus to follow in the footsteps of Hartline and Roughgarden [2008] and the revenue maximization literature, and attempt to find a simple mechanism that can approximate the utility of the optimal mechanism.

**Main Contribution: Generalizing Hartline and Roughgarden [2008] to the Multidimensional Setting.** The main focus of this paper is generalizing the results of Hartline and Roughgarden [2008] from the single-dimensional $m$ identical items setting to the multidimensional $m$ i.i.d. items setting for unit-demand bidders. (See Remark 1.) That is, we aim to answer the following question.

**Question 1.** In the i.i.d. unit-demand setting with $n$ buyers and $m$ items, what utility can we obtain in approximation to optimal social welfare and by what simple mechanisms?

We do exactly this, tackling the complexities that arise from the multidimensional setting, designing simple mechanisms, and matching the tight single-dimensional bounds. We use a class of mechanisms that we refer to as Favorites Mechanisms, further discussed in Section 3.1.

*Favorites Mechanisms:* We use the same underlying principle for all of our results: each buyer points to their favorite item, and then allocation is handled per-item among those who declare it their favorite via some fixed single-item mechanism. In the more-items-than-bidders regime, the single-item mechanism assigns items uniformly at random among those competing for the same favorite. In the more-bidders-than-items regime, we use a known single-dimensional prior-free mechanism from Hartline and Roughgarden [2008] to allocate each individual item.

*Results.* In this paper, we obtain utility that approximates optimal welfare when there are $n$ buyers, $m$ items, and buyer values for items are drawn i.i.d. across both items and buyers. We achieve a $(1 - 1/e)$-approximation when there are more items than buyers ($m \geq n$) and an $O(\log \frac{n}{m})$-approximation when there are more buyers than items ($n > m$). We then create an instance of the identical-item setting from the i.i.d. unit-demand setting without increasing social welfare too much and only increasing optimal utility, allowing us to leverage the tight example from Hartline and Roughgarden [2008]. All together, Section 3 answers Question 1 with a $\Theta(1 + \log \frac{n}{m})$ gap between optimal utility and social welfare.

**Additional Contribution: Discussion on the Complexity of Multidimensional Bayesian Utility Maximization.** A natural first approach in studying this problem would be to attempt to leverage the tools from the well-studied field of revenue maximization. We formalize the intuition behind why some of these powerful techniques cannot be naively applied in this setting. Our second contribution is an extensive discussion on general multidimensional Bayesian utility maximization, why this direction is likely to be technically interesting and challenging, and how it differs from what is known about multidimensional Bayesian revenue maximization (Section 4 and Appendix D).

The tightness of our main contribution demonstrates a need for better upper bounds than welfare. While revenue maximization does offer tools for developing such bounds, we face two key hurdles. First, we show that substituting a bidder with multiple single-minded copies of themselves—while keeping the total number of items fixed—can actually reduce utility. This implies that the widely used "copies" technique from revenue maximization fails to provide a universal upper bound (see Theorem 4). Second, known duality techniques reduce the problem to constructing a dual for the single-bidder problem, but as mentioned above, the single-bidder multidimensional problem is trivial and provides no insight into the structure of the optimal multidimensional mechanism. Moreover, attempts to adapt the successful flows from revenue maximization can yield bounds even weaker than welfare, highlighting the challenge of applying these techniques in our setting.

Going beyond revenue maximization, we then check whether a key insight from the single-item utility maximization literature still holds. Namely, in the single-item setting, the designer can always allocate the item to some bidder and need not consider the outcome where the item is left unallocated. However, we find that this is no longer true in the multidimensional setting. Instead, the *utility*-optimal mechanism sometimes must *throw away* an item just to guarantee optimality (Theorem 6)! Of course, we see similar mechanisms in revenue maximization, but there the goal is to maximize payments at any cost. Here, the objective is to simultaneously allocate efficiently and minimize payments, so the idea that *not allocating* could somehow help defies intuition. But in the multidimensional setting, depending on how strong a buyer's preference for one item over another is, not allocating can actually reduce payments enough to improve utility overall (Observation 1).

By demonstrating why standard approaches from revenue maximization fail in this setting, we expose key gaps in our understanding, and shed light on the new technical challenges posed by Bayesian utility maximization, which we anticipate will generate significant interest for future work. We hope that our discussion serves as a roadmap for researchers entering this area, offering intuition, highlighting key obstacles, and identifying interesting directions for future work.

## 1.1 Related Work

While well-explored in the public finance literature in economics and the strategic queuing literature in operations, utility maximization and ordeals have minimal prior work from the algorithmic mechanism design perspective.

In 2008, Hartline and Roughgarden completely resolved the question of optimal utility maximization in the single-dimensional Bayesian setting. Hartline and Roughgarden [2008] also introduce an idea for a single-dimensional prior-free benchmark and provide a constant-approximation to that benchmark, in addition to their tight $\Theta(1 + \log \frac{n}{m})$-approximation to social welfare.

The best (and only) known multidimensional approximation prior to our work is $O(\log |\Omega|)$ where $\Omega$ is the space of possible allocation outcomes [Fotakis et al., 2016]. In the unit-demand setting, this is $O(m \log n)$. The mechanism used is a combination of VCG and the uniformly random free allocation that uses ideas from the above prior-free mechanism. We discuss in Section 3.1 why these ideas are not enough, and how the idea of Favorites is crucial. In addition, Fotakis et al. [2016] give an algorithmic non-constructive result that achieves the same utility approximation while simultaneously guaranteeing constant welfare, although they note that running VCG with probability 1/2 will do this as well.

Very recent subsequent work by Ezra et al. [2025] builds on our work in the i.i.d. setting to also designs mechanisms whose utility approximates the benchmark of optimal social welfare. Most salient, they achieve a general $O(1 + \log \frac{cn}{m})$-approximation for the condition that the probability of any item being an agent's favorite is at most $c/m$, characterizing the improvement of utility in $m$ and $c$. This implies a $\Theta(\log n)$-gap between optimal utility and social welfare for general unit-demand bidders and matches our $O(\log \frac{n}{m})$-approximation for the i.i.d. unit-demand setting (and in fact, for the uniform favorites setting of Appendix C). A deeper comparison of their work and ours can be found in Appendix A.

A much more in-depth review of related work in fields adjacent to computer science, related but adjacent settings, and more domain-specific work is provided in Appendix A.

*This work.* The goal of this work is to take the simple-yet-approximately-optimal approach (reviewed further in Appendix A) for the setting of multidimensional Bayesian utility maximization, and it is the *first* paper to do so. In particular, we focus on what the right simple mechanisms may be for this objective, approximate the benchmark of welfare, and study the i.i.d. unit-demand setting.

## 2 Preliminaries

We consider the setting where the mechanism designer wishes to sell $m$ heterogeneous items to $n$ bidders. Agent $i$'s valuation is a vector $v_i \in \mathbb{R}^m$, where $v_{ij}$ denotes agents $i$'s private value (or willingness-to-pay) for item $j$. Value $v_{ij}$ is drawn independently from a known prior distribution $v_{ij} \sim F_{ij}$, and in particular, we assume that values are independently and identically distributed (i.i.d.) across both items and bidders. That is, that there exists some type distribution $F$ such that

$F_{ij} = F$ for all $i, j$. We overload notation and let $F(\cdot)$ denote the CDF and $f(\cdot)$ denote the PDF of the distribution. Let $\mathbf{F} = F^{n \times m}$ denote the joint distribution. We use $\mathbf{v}$ to denote the vector of all agents' types, $\mathbf{v} = (v_1, \ldots, v_n) \in \mathbb{R}^{n \times m}$.

**Remark 1** (I.I.D. Unit-Demand). We generalize the "multi-unit auction" setting of Hartline and Roughgarden [2008], where there are $m$ identical items, and each agent has a single value for all $m$ items. Our extension instead considers when distinct item values are drawn independently and identically distributed (i.i.d.), with buyers remaining unit-demand. That is, the items are non-identical, heterogeneous. As with when distributional assumptions are used in Hartline and Roughgarden [2008], values are also i.i.d. across buyers.

A mechanism $\mathcal{M}$ is comprised of an allocation rule $\mathbf{x} : \mathbb{R}^{n \times m} \to [0, 1]^{n \times m}$ and a payment rule $\mathbf{p} : \mathbb{R}^{n \times m} \to \mathbb{R}^n$ where $x_i(\mathbf{v})$ and $p_i(\mathbf{v})$ are the respective allocation and payment for agent $i$ under type $\mathbf{v}$. Note that $x_i(\mathbf{v})$ is a vector where $x_{ij}(\mathbf{v})$ denotes the probability that bidder $i$ receives item $j$ under type profile $\mathbf{v}$, and $p_i(\mathbf{v})$ is a non-negative number.

Agents are *unit-demand*, wanting at most one item. Formally, we would write $v_i(S) = \max_{j \in S} v_{ij}$; agent $i$ only gets value from his favorite item among those allocated. However, it is without loss to restrict attention to allocation rules that allocate at most one item to each agent, that is, have the feasibility constraint $\sum_j x_{ij}(\mathbf{v}) \leq 1$ for all agents $i$ and all $\mathbf{v}$. Then agent $i$'s expected utility is $\sum_j x_{ij}(\mathbf{v})v_{ij} - p_i$, which we rewrite as $x_i(\mathbf{v})v_i - p_i(\mathbf{v})$, where the first term represents the dot product of two vectors.

We define the expected social welfare, utility, and revenue objectives: $\text{UTILITY} = \mathbb{E}_{\mathbf{v} \sim \mathbf{F}} \left[ \sum_i x_i(\mathbf{v})v_i - p_i(\mathbf{v}) \right], \text{WELFARE} = \mathbb{E}_{\mathbf{v} \sim \mathbf{F}} \left[ \sum_i x_i(\mathbf{v})v_i \right]$, and $\text{REVENUE} = \mathbb{E}_{\mathbf{v} \sim \mathbf{F}} \left[ \sum_i p_i(\mathbf{v}) \right]$ and observe by linearity of expectation that $\text{UTILITY} = \text{WELFARE} - \text{REVENUE}$.

A mechanism is Bayesian incentive-compatible (BIC) if, in expectation over the other agents' reports, it incentivizes truthful reporting:

$$\mathbb{E}_{\mathbf{v}_{-i} \sim \mathbf{F}_{-i}} \left[ x_i(v_i, \mathbf{v}_{-i})v_i - p_i(v_i, \mathbf{v}_{-i}) \right] \geq \mathbb{E}_{\mathbf{v}_{-i} \sim \mathbf{F}_{-i}} \left[ x_i(v_i', \mathbf{v}_{-i})v_i - p_i(v_i', \mathbf{v}_{-i}) \right] \quad \forall i, v_i, v_i'$$

where the subscript $-i$ indexes the vector everywhere but $i$.

For an objective, we let OPT denote the optimal amount attainable by any feasible, truthful mechanism, that is, $\text{OPT}_{\text{UTILITY}} = \max_{(\mathbf{x}, \mathbf{p}) \in \mathcal{P}} \mathbb{E}_{\mathbf{v} \sim \mathbf{F}} \left[ \sum_i x_i(\mathbf{v})v_i - p_i(\mathbf{v}) \right]$, where $\mathcal{P}$ denotes the set of feasible, BIC, and individually rational mechanisms. Namely for feasibility $\sum_j x_{ij}(\mathbf{v}) \leq 1 \quad \forall i, \mathbf{v}$ and $\sum_i x_{ij}(\mathbf{v}) \leq 1 \quad \forall j, \mathbf{v}$ and for individual rationality $\mathbb{E}_{\mathbf{v}_{-i} \sim \mathbf{F}_{-i}} \left[ x_i(v_i, \mathbf{v}_{-i})v_i - p_i(v_i, \mathbf{v}_{-i}) \right] \geq 0 \quad \forall i, v_i$.

# 3  Main Result: Approximating Utility via Welfare

In this section, we generalize the seminal result of Hartline and Roughgarden [2008]: we prove a bound on the gap between the optimal welfare and the optimal utility in the i.i.d. unit-demand setting. They study the $m$ identical item setting with $n$ i.i.d agents and construct a prior-free mechanism that achieves a tight bound on this gap. We generalize this result to the multidimensional setting, when the $n$ i.i.d. buyers have *identically distributed* values for the $m$ items. This introduces all of the complexities of the multidimensional setting, which we elaborate on in Section 4 and Appendix D.

The best known bound on this gap is that optimal utility is a $\log(|\Omega|)$-approximation to optimal welfare [Fotakis et al., 2016], where $\Omega$ is the set of possible outcomes. With $n$ agents and $m$ items the number of possible outcomes is $n^m$, meaning this bound is approximately $O(m \log n)$.

Our results give two new tighter bounds for the i.i.d. unit-demand setting: in the case where $m \geq n$, we give a constant $1 - \frac{1}{e}$ bound, and in the case where $m < n$, we give an $O(\log \frac{n}{m})$ bound, which we then prove is tight up to constant factors, implying an overall $\Theta(1 + \log \frac{n}{m})$-bound.

We use the $\mathcal{M}$-Favorites mechanism for both cases: separating bidders by their favorite item into a single-item setting. However, we use different single-dimensional mechanisms $\mathcal{M}$ in each regime to allocate the items: Random-Favorites in the $m \geq n$ regime and Prior-Free-Favorites when $n > m$.

It may appear that these regimes could be unified via the Prior-Free-Favorites approach. We point out in Section 3.3 where the analysis breaks for $n < m$, rendering both regimes necessary.

We remark that neither mechanism requires knowledge of the prior distribution, so these mechanisms are *prior-independent*. However, our attention is restricted to the setting where values are i.i.d. across both buyers and items in order for BIC and the approximation analyses to hold.[3]

## 3.1 Candidate Simple Mechanisms

In this section, we motivate candidate simple mechanisms. We seek an analogue to those in the revenue literature, where selling separately and selling the grand bundle are the two simple mechanisms that, together (and with an entry-fee), capture all cases and give a constant-factor approximation. We start by considering the optimal single-dimensional mechanisms from Hartline and Roughgarden [2008] and their potential multidimensional analogues.

At its extremes, the utility-optimal single-dimensional mechanism, discussed further in Appendix D.1, looks either like a uniformly random free allocation or a second-price auction. For this reason, we take the multidimensional analogues of these two mechanisms as candidates for natural "simple" mechanisms that we may wish to consider when approximating utility—(1) a uniformly random free allocation of goods to buyers, and (2) the VCG mechanism [Vickrey, 1961, Clarke, 1971, Groves, 1973], the welfare-optimal mechanism that is the multidimensional analogue of the second-price auction.

Randomly giving the items away for free has clear utility benefits, as it is the payment-minimizing allocation. However, with multiple items, it is easy to imagine an outcome where two of the random winners would prefer to exchange items. Then a clear improvement over this mechanism would allow agents to express preferences over which item they are in contention for.

In contrast, VCG not only allows agents to express preferences, but it ensures that the items are allocated in an efficient way. When agents prefer different items, VCG can actually allocate these items for free, yielding utility that matches welfare. Where VCG does poorly is when multiple bidders prefer the same item, VCG charges potentially very large payments in order to ensure the efficient allocation. In fact, Fotakis et al. [2016] show that VCG's utility cannot better than $|\Omega|$-approximate social welfare where $\Omega$ is the space of allocative outcomes, but their proof uses single-minded agents (with complementarities) to show this.

This motivates a mechanism that allows bidders to express their preferred item while handling "collisions" in buyer preferences in a way that does not hurt utility too much. Our simple mechanism essentially must be of the form where buyers express their favorite item, and allocation is determined from there. We define a family of mechanisms which we call $\mathcal{M}$-Favorites (Definition 1), which allow agents to report their favorite items, and then uses some fixed mechanism $\mathcal{M}$ to resolve conflicts between bidders with the same favorite. A huge added benefit is the tractability this brings by allowing us to decompose the problem into many single-item problems. Moreover, we have the property that if items are distributed i.i.d., then $\mathcal{M}$-Favorites is BIC whenever $\mathcal{M}$ is BIC (Lemma 1).

**Definition 1** ($\mathcal{M}$-Favorites)**.** Each bidder announces their favorite item $j$, the item for which they drew the highest value. Then, for the given BIC single-item mechanism $\mathcal{M}$, for each item $j$, $\mathcal{M}$ is run on the bidders who reported item $j$ as their favorite.

In particular, we will focus on two variants.

- Random-Favorites: $\mathcal{M}$ allocates uniformly at random among all participants (and charges nothing). That is, each item is allocated uniformly at random among one of the bidders who declares it their favorite.

- Prior-Free-Favorites: $\mathcal{M}$ is the single-item prior-free mechanism below.

**Definition 2** (Single-Item Prior-Free Mechanism)**.** The mechanism of Hartline and Roughgarden [2008] used as a subroutine in our multidimensional Prior-Free-Favorites mechanism is as follows: Rename the bids participating in this single-dimensional mechanism $v_1 > \ldots > v_{n'}$. Choose $j$ uniformly at random from $\{0, \ldots, \log(n')\}$. Run a $v_{2^j+1}$-lottery (with $v_{n'+1} := 0$): choose a bidder uniformly at random from those who report at least $v_{2^j+1}$, then allocate to this bidder and charge

---

[3]The majority of multidimensional prior-independent mechanisms require that values be drawn i.i.d. across bidders, but do not require this across items. See [Deng and Zhang, 2021, Goldner and Karlin, 2016, Deng et al., 2022] for additional work on prior-independent mechanisms. We discuss the limitation of our approach beyond i.i.d. in Appendix C.

price $v_{2^j+1}$. That is, the bidders are ordered by report, a power-of-two bidder is chosen uniformly at random, and then this bidder is used as a price, and the mechanism allocates to a bidder chosen uniformly at random from those who reported above this price.

We now prove that this mechanism is truthful in the i.i.d. setting.

**Lemma 1.** *The $\mathcal{M}$-Favorites mechanism is Bayesian incentive-compatible (BIC).*

*Proof.* Since the single-item subroutine $\mathcal{M}$ is BIC, its interim allocation rule for any agent $i$ can only depend on the expected number of participants in the mechanism, their prior distributions, and $i$'s report. We now argue that for every agent $i$, these parameters are the same for every item $j$, hence it is in every agent's best interest to report their favorite item, and then report their true value to the already BIC single-item subroutine.

As each bidder's value for each item $v_{ij}$ is drawn i.i.d, the same expected number of bidders will report each item as their favorite, hence each item will have the same expected number of participants in its single-item mechanism. The i.i.d. assumption also guarantees that the mechanism's input prior distributions are the same. Since $\mathcal{M}$ is truthful, the allocation obtained by participating in it is monotone non-decreasing in the bidder's report. Hence the bidder will achieve the highest utility by participating in the mechanism for the item for which they have the highest value: their favorite item. Since this allocation is BIC,[4] there exist payments that implement it.  $\square$

While the mechanism of Ezra et al. [2025] is too complex to describe in full here, at a high-level, it creates $2^\ell$ copies of every item for a randomly drawn $\ell$, runs VCG to determine allocations and prices, and then randomly determines which of the copies is real. This is far more complex and less practical to implement than the $\mathcal{M}$-favorites mechanism.

Having motivated the use of $\mathcal{M}$-Favorites to approximate expected utility via social welfare in the i.i.d. setting, we are ready for our main results.

### 3.2  More Items than Bidders

In this section we prove a constant bound between optimal social welfare and optimal utility using the Random-Favorites mechanism.

**Theorem 1.** *Random-Favorites achieves utility at least $\left(1 - \frac{1}{e}\right) \mathrm{OPT_{WELFARE}}$ when $m \geq n$.*

*Proof.* We begin with the case of an equal number of bidders and items ($m = n$). In the Random-Favorites mechanism, an item is *not* allocated only when it is not any agent's favorite item, which occurs with probability $(1 - 1/m)^n$. As $m = n$, then the item *is* allocated with probability $1 - \left(1 - \frac{1}{m}\right)^n \geq 1 - \frac{1}{e}$. Because $m = n$, if an item goes unallocated, then so does an agent. Hence, the probability that an item is allocated is equal to the probability that an agent is allocated.

As $m$ grows larger than $n$, the probability that an *agent* is allocated only increases: with more items, the probability that agents "collide," or share their favorite item, is less likely, and this is the only event in which an agent is not allocated. Then for all $m \geq n$, the probability that an agent is allocated in the Random-Favorites mechanism is at least $1 - 1/e$.

An agent's value for their favorite item is, by definition, at least as high as their value for the item they receive in the social-welfare-optimal allocation. Hence we guarantee utility that is a $1 - \frac{1}{e}$ fraction of the optimal welfare.  $\square$

### 3.3  More Bidders than Items

In this section, we prove a logarithmic bound between utility and social welfare in the case where there are more bidders than items, i.e., $m < n$. We do this using the Prior-Free-Favorites mechanism, in which bidders declare their favorite item, and are then divided into $m$ single-item auctions based on what they declared. The single-item prior-free mechanism is then used as a subroutine to allocate each item $j$ amongst those who declare $j$ their favorite item.

---

[4]Note that this procedure is not DSIC: the allocation is only the same across items in expectation.

The proof follows three steps. First, we show that for any item $j$ that the gap between the utility achieved and the highest value amongst all bidders whose favorite item is $j$ is logarithmic in the number of bidders who have that item as their favorite. Second, we lower bound the probability that the bidder with the highest value for $j$ among all bidders has $j$ as their favorite item. Then, we combine these facts using a technical lemma (Lemma 3) to prove the claim.

**Theorem 2.** *In the i.i.d. unit-demand setting with $n$ bidders and $m$ items, when $n > m$, the Prior-Free-Favorites mechanism achieves utility that $O(\log \frac{n}{m})$-approximates optimal social welfare.*

*Proof.* Consider any item $j$. Suppose there are $n'$ bidders who reported item $j$ as their favorite item, that is, $n'$ bidders participate in the mechanism for $j$. Without loss of generality, we order the bidders by their values $v_1 \geq v_2 \ldots \geq v_{n'}$. Within the single-item setting for item $j$ that the Favorites mechanism has created, Theorem 5.2 from Hartline and Roughgarden [2008] shows that the single-item prior-free mechanism, our subroutine, guarantees utility at least $\frac{v_1}{2(1+\log_2(n'))}$.

We now state and prove a lemma lower-bounding the probability that "favorites align" for any item.

**Lemma 2.** *Let* favorites align *for item $j$ be the event that the agent with the highest value for item $j$ among all agents ($j$'s favorite bidder) prefers item $j$. When $n > m$, this occurs for each item $j$ independently probability at least $\frac{1}{2}$.*

*Proof.* Consider the agent $i$ with the highest value $v_{ij}$ for $j$ among *all* agents. We now bound the probability that $v_1$, the highest value competing for $j$, is not $v_{ij}$. This occurs when agent $i$ is competing for a different favorite item $k$, that is, $v_{ik} > v_{ij}$.

What is the probability that $i$ prefers some other item $k$, that is, $v_{ik} > v_{ij}$? Consider $n + m - 1$ draws from the type distribution: $n$ draws for each bidder's value for item $j$, and $m - 1$ draws for $v_{ik}$ for all $k \neq j$. Label $m - 1$ of these draws as the $v_{ik}$'s uniformly at random. Of the remaining $n$ draws for bidders' values for item $j$, $v_{ij}$ must be the largest, by definition. Using this sampling process, as $n > m$, the probability that some $v_{ik} > v_{ij}$ is $< \frac{1}{2}$, hence $\Pr[v_{ik} \leq v_{ij}] \geq \frac{1}{2}$. $\square$

By Lemma 2, for each item $j$, favorites align with probability at least $\frac{1}{2}$, and thus the agent with the highest value for item $j$ is competing for item $j$ in the Favorites mechanism. Therefore, per item $j$, we yield expected utility at least $\mathbb{E}_{n'}\left[\frac{1}{2} \cdot \frac{v_{ij}}{2(1+\log(n'))}\right]$.

We now combine the guarantees from the single-item subroutines with the probability that favorites align to show that our multidimensional mechanism gives our guarantee. We use a technical lemma (Lemma 3) that shows that the correlation between the largest value for $j$, $v_{ij}$, and the number of bidders who prefer item $j$, $n'$, is well-enough behaved (i.e. $v_{ij}$ is not only large when $n'$ is large, or high-valued bidders do not only participate for competitive items). The lemma is stated in slightly more general language of probability distributions, which we then translate back to our setting.

**Lemma 3.** *Suppose $V$ is the max of $n'$ draws from the distribution of the random variable $X$. $n'$ is distributed according to Binomial$(n, 1/m)$. For some $v'$, $\Pr[V \geq v'] \geq 1/2$. Then for a constant $c = 1/4.45$:*

$$\mathbb{E}\left[\frac{V}{1 + \log n'} \mid V \geq v'\right] \geq c \cdot \frac{\mathbb{E}[V]}{1 + \log \mathbb{E}[n']}.$$

In our case, $V = v_{ij}$ is the max of $n'$ draws from the i.i.d. distribution for values for item $j$, where each bidder has $j$ as their favorite item with success probability $1/m$, and $\Pr[V \geq v_{ik}] \geq 1/2$ (per Lemma 2). Then by Lemma 3, per item $j$, we yield expected utility at least

$$\mathbb{E}_{n'}\left[\frac{1}{4} \cdot \frac{v_{ij}}{1 + \log(n')} \mid v_{ij} \geq v_{ik}\right] \geq \frac{1}{4.45} \cdot \frac{1}{4} \cdot \frac{\mathbb{E}_{n'}[v_{ij}]}{1 + \log(\mathbb{E}[n'])}.$$

Since values are i.i.d., $\mathbb{E}_{n'}[n'] = \frac{n}{m}$, so the per-item utility is at least $\frac{v_{ij}}{17.8(1+\log \frac{n}{m})}$. Summing across items and using the fact that $v_{ij} = v_1$, we get that the expected utility of Prior-Free-Favorites is at least a $\log$-fraction of the optimal social welfare:

$$\mathbb{E}[\text{UTILITY}] \geq \frac{\text{OPT}_{\text{WELFARE}}}{17.8(1 + \log(\frac{n}{m}))}. \qquad \square$$

**Remark 2.** Note that this analysis cannot to be directly used to also give a constant approximation when $n < m$—that is, we need our separate analysis of the $n < m$ regime. This analysis uses a bound on the probability that an item's "favorite" agent is also that agent's favorite item. When the number of *items* grows large, this probability tends to $0$.

## 3.4 Tightness

In this section, we prove that our approximation for the $n > m$ regime is tight up to constant factors. Since we obtain a constant-factor approximation in the $n \leq m$ regime, then together, this gives a tight $\Theta(1 + \log \frac{n}{m})$ gap between optimal utility and social welfare.

Proposition 5.1 in Hartline and Roughgarden [2008] shows a matching lower-bound of $\Omega(1 + \log \frac{n}{m})$ for an instance with $m$ identical items. Note, however, that this instance is not contained within the setting of $m$ i.i.d. items, so it does not immediately apply to our setting. One could imagine that in the i.i.d. setting, each agent having additional independent draws from the value distribution could yield a strictly better ratio of optimal welfare to utility than in the identical-item case.

However, in this setting, we show that our approximation is indeed tight (Theorem 3) using this existing lower bound. We transform the i.i.d. unit-demand setting into the identical item setting (Definition 3), only increasing optimal utility (Lemma 4), and not increasing optimal welfare too much (Lemma 5). Together, this allows us to translate the lower-bound from the identical setting into one for the i.i.d. unit-demand setting for every $m$ and $n$.

**Definition 3.** Given an instance $I$ with $n$ bidders, $m$ items, and all values drawn i.i.d. from a distribution $F$, let $I'$ be the instance where for each bidder $i$, all $m$ of $i$'s item values are replaced with the maximum of bidder $i$'s values in instance $I$. That is, $(v_{ij})^{I'} = \max_j (v_{ij})^I \quad \forall i$. Observe that this transforms the i.i.d. unit-demand setting into a setting with $m$ identical items where values are now only i.i.d. across bidders.

**Theorem 3.** *In the i.i.d. unit-demand setting, the gap between optimal utility and optimal social welfare is* $\text{OPT}_{\text{WELFARE}}/\text{OPT}_{\text{UTILITY}} = \Theta(1 + \log \frac{n}{m})$.

*Proof.* We define an i.i.d. unit-demand instance $I$ on $n$ bidders $m$ items, where all values are drawn i.i.d. from a distribution $F$ such that the maximum of $m$ draws from $F$ is exponential(1). Using Definition 3, we then construct a corresponding identical-items instance $I'$.

By Lemma 4, the utility of the identical-items instance is only larger: $\text{OPT}_{\text{UTILITY}}(I') \geq \text{OPT}_{\text{UTILITY}}(I)$. By Lemma 5, the welfare of the identical-items instance is within a constant of the original i.i.d. unit-demand instance: $c \cdot \text{OPT}_{\text{WELFARE}}(I) \geq \text{OPT}_{\text{WELFARE}}(I')$ for some constant $c$.

By Definition 3's construction of an identical-items instance from $I$, the distribution used in our construction of instance $I$, and Proposition 5.1 of Hartline and Roughgarden [2008] on $I'$, $\text{OPT}_{\text{WELFARE}}(I') = \Theta(m(1 + \log(\frac{n}{m}))$ and $\text{OPT}_{\text{UTILITY}}(I') \leq m$. As $I$ is an instance in the i.i.d. unit-demand setting, the theorem holds. $\qquad\square$

**Lemma 4.** *For any instance $I$ and the transformation to $I'$ in Definition 3, $\text{OPT}_{\text{UTILITY}}(I') \geq \text{OPT}_{\text{UTILITY}}(I)$.*

*Proof.* Our proof consists of three immediate observations. First, for $m$-item setting, it does not decrease the optimal utility to constrain attention to mechanisms that only allocate $m$ items, as there were only $m$ items to allocate. The original utility-optimal mechanism is still valid.

Second, it cannot not decrease optimal utility to create $n$ copies of every item: there is now only less competition amongst bidders, which improves utility. Hence we consider the setting where there are $m \cdot n$ items ($n$ copies of each different item), but only $m$ total items can be allocated.

Third, in this setting, in the utility-optimal mechanism (that only allocates $m$ items) always, without loss, each bidder that gets allocated receives their favorite item: they have more value for their favorite item than any other item, and one of the $n$ copies is available without competition. This mechanism gets at least as much utility as the original mechanism which allocates $m$ different items to agents, as the allocated agents now have value at least as high for their received items.

This new setting has optimal utility equal to that in the instance $I'$, as each agent's only competitive value in the mechanism is their maximum value. Hence, the utility-optimal mechanism with $n$

copies of each item that allocates at most $m$ items obtains utility equal to the instance $I'$, and this is only more than the optimal utility for the original instance $I$. $\qquad\square$

**Lemma 5.** *For an instance $I$ with $n$ bidders, $m$ items, and all values drawn i.i.d. from a distribution $F$ such that the maximum of $m$ draws from $F$ is exponential(1), and for the transformation to $I'$ in Definition 3, then for some constant $c$, $c \cdot \text{OPT}_{\text{WELFARE}}(I) \geq \text{OPT}_{\text{WELFARE}}(I')$.*

*Proof.* First, we consider the upper bound on social welfare that could be achieved if we didn't need to respect unit-demand constraints. That is, we could just give each of $m$ items to the bidder with the highest value for it, which is the maximum draw over $n$ bidders. Denote $X_n^1$ as the maximum of $n$ draws of the random variable $X$ drawn from distribution $F$. Then this quantity is $m \cdot \mathbb{E}[X_n^1]$. We show that this is actually a constant approximation to social welfare.

Consider the event where favorites are aligned for item $j$: the maximum value for item $j$ belongs to a buyer who prefers item $j$. By Lemma 2, this event occurs with probability at least $\frac{1}{2}$. When this event occurs, it is feasible even with unit-demand constraints to allocate item $j$ to the bidder that values it the most. Finally, notice that conditioning on the event of favorites aligned means that the maximum draw for the item is also larger than all of the buyer's other values, but this is only larger than not conditioning on this: $\mathbb{E}[X_n^1 \mid \text{favorites aligned}] \geq \mathbb{E}[X_n^1]$. By summing the contribution of each item $j$ in the event that it has favorites aligned, we achieve $\text{OPT}_{\text{WELFARE}}(I) \geq \frac{1}{2} m \cdot \mathbb{E}[X_n^1]$.

Next, recall that by definition of instance $I$, when $X$ is drawn from $F$, $X_m^1$ is distributed like an exponential(1) random variable. Denote $Y$ as a random variable drawn from exponential(1). Then as $n > m$, if $n/m$ is an integer, we can divide the $n$ draws into $n/m$ groups of $m$ draws, take the maximum of each group first, and then the maximum of the maximums to achieve $X_n^1$, hence $\mathbb{E}[X_n^1] \geq \mathbb{E}[Y_{\lfloor \frac{n}{m} \rfloor}^1]$. That is, for any (potentially non-integral) $n/m$, $X_n^1$ is at least the maximum of $\lfloor \frac{n}{m} \rfloor$ draws from an exponential(1) distribution. Using an identity of exponential distributions, we get $\mathbb{E}[X_n^1] \geq \mathbb{E}[Y_{\lfloor \frac{n}{m} \rfloor}^1] \geq 0.5 + \max\{0, \log\left(\frac{n}{m}\right)\}$. Putting these steps together, we obtain $\text{OPT}_{\text{WELFARE}}(I) \geq \frac{1}{2} m \cdot \left(0.5 + \max\{0, \log\left(\frac{n}{m}\right)\}\right)$.

We now compute $\text{OPT}_{\text{WELFARE}}(I')$. Using our definition of $I$ and the transformation in Definition 3, $I'$ is equivalent to an instance in which there are $m$ identical copies of a good, and each bidder's value is drawn from exponential(1). Proposition 5.1 of Hartline and Roughgarden [2008] then gives that the welfare of $I'$ is $\Theta(m(1 + \log(\frac{n}{m})))$.

Finally, $\frac{\text{OPT}_{\text{WELFARE}}(I')}{\text{OPT}_{\text{WELFARE}}(I)} \leq \frac{m(1 + \log(\frac{n}{m}))}{\frac{1}{2} m \cdot \left(0.5 + \max\{0, \log(\frac{n}{m})\}\right)} \leq \frac{2(1 + \log(\frac{n}{m}))}{0.5 + \max\{0, \log\left(\frac{n}{m}\right)\}}$, which is a constant for all $\frac{n}{m} \geq 1$. $\qquad\square$

# 4  Discussion: Finding An Upper Bound on Optimal Utility

In this section, we also highlight difficulties in obtaining a better upper bound on optimal utility than social welfare, despite the arsenal of techniques from the revenue literature. Further discussion regarding the complexities of the general unit-demand setting can be found in Appendix D.

Approximation to an unknown quantity like optimal expected utility requires a good (small) upper bound. The most naive upper bound on utility is optimal *social welfare*, which is what our result approximates, and thus also bounds the gap between optimal utility and optimal welfare in the Bayesian setting.

However, one might wonder if there's a nice upper bound on optimal utility that is tighter than welfare. Our answer is, perhaps, but certainly not one that is easy to find. For instance, the revenue maximization literature provides many different approaches, but to the best of our knowledge, none of them can easily be adapted to utility to obtain a better upper bound. To illustrate this point, we briefly discuss the difficulties in using two of these upper bound approaches: the copies setting and the Lagrangian duality flow-based upper bound.

**The Copies Setting.** A commonly used upper bound on revenue is the copies setting, first introduced in Chawla et al. [2007]. Each multidimensional bidder is split into $m$ copies who possess the same value as the original agent for item $j$, but are *only* interested in item $j$. This is now a single-dimensional setting. The copies setting is quite useful for bounding optimal revenue; for instance,

the revenue of every deterministic unit-demand mechanism is upper-bounded by the optimal (single-dimensional) revenue in the copies setting [Chawla et al., 2010]. However, this is not the case for utility. While splitting each bidder into copies eliminates incentive issues across items, it also increases the competition for each item. Further, these copy-bidders are single-minded, and thus their demands cannot be balanced across items. This form of increased competition drives down utility. We formalize this intuition below.

**Theorem 4.** *There exists a distribution $F$ and deterministic mechanism $M$ such that the expected utility of $M$ over $F$ is greater than the single-item optimal utility in the copies setting induced by $F$.*

*Proof.* Consider a setting with 2 items and 2 bidders whose values are drawn i.i.d. from any anti-MHR distribution. The induced copies setting includes a constraint that only one of the copies of each bidder can be allocated. Clearly relaxing this constraint can only increase the utility that can be achieved. In this relaxed copies setting, because $F$ is anti-MHR and there are no constraints across items, the optimal per-item mechanism is a Vickrey (second-price) auction.

In the VCG mechanism, for each value profile in which both bidders have the same favorite item, VCG achieves the same utility as the relaxed copies setting. However, when the bidders have different favorite items, VCG allocates efficiently while charging a payment of 0, while the copies setting must still charge a price, resulting in less utility. □

**Remark 3** (Item-Copies). Note that the gap between optimal utility and optimal social welfare in the general unit-demand setting is $\Theta(\log n)$. This is because for general distributions, for any number of items, the single-item case is still contained. For example, if all item distributions aside from one item are a point mass as 0, this matches the $\Omega(\log n)$ gap from the single-item setting.

One might consider a variant on the copies setting that instead splits each item into multiple copies, as in the mechanism from Ezra et al. [2025]. However, the above pathological example also shows that $n$ copies of that item would be required in order for, e.g., the utility of the VCG mechanism in the item-copies setting to produce an upper bound on optimal utility. But then, of course, the optimal utility of the setting with $n$ copies is just social welfare.

**Duality-Based Upper Bounds** Another technique used with great success in recent years is a Lagrangian-duality-based approach to formulating an upper bound on optimal revenue. This method formulates the optimization problem as a linear program, relaxes certain constraints using Lagrange multipliers, and takes its dual program. Within the dual program, any feasible dual variables result in a dual objective which upper bounds the optimal primal. Cai et al. [2016] reinterpret these feasibility constraints as a network flow problem, and use this structure to construct dual variables that lead to a nice (tighter) upper bound on revenue.

While the dual of the utility maximization linear program can also be expressed as a network flow problem, it lacks some of the key properties used in the revenue maximization approach to construct the "right" dual variables. Importantly, the single-bidder revenue maximization flow problem has an optimal solution that can be generalized to the multi-bidder setting. The single-bidder utility-maximization dual, however, has countless trivial optimal solutions to the flow problem. Furthermore, some of these solutions give the trivial bound of welfare when extended beyond the single-bidder setting. Then to create a useful flow, one must work in the much more complex multi-bidder setting at the onset. Furthermore, analogues of and variations on the "canonical" Cai et al. [2016] multi-item revenue flow yield upper bounds for utility that are even greater than social welfare.

**An Open Question.** The above motivates a very important and large open question for this community, likely to require technical innovation: what is a tractable upper bound on optimal utility? Is there a framework in the spirit of [Cai et al., 2016] to produce such upper bounds across various multidimensional environments?

## Acknowledgments

The idea for this research direction originated jointly with Anna Karlin out of conversations with Mark Braverman about [Braverman et al., 2016]. We would like to thank Anna Karlin, Matt Weinberg, and Alon Eden for helpful conversations in early stages of this work. We also thank Nathan Klein and Divyarthi Mohan for key ideas in the proof of Lemma 3.

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

# A    Expanded Related Work

In this appendix, we expand on the related work in Section 1.1 that was abbreviated due to space limitations. Please see Section 1.1 for the fundamental computer science literature, namely Hartline and Roughgarden [2008], Fotakis et al. [2016], Ezra et al. [2025].

Utility maximization has been well-explored from the public finance literature in economics [McGee, 2003] and the strategic queuing literature in operations [Hassin and Haviv, 2003]. Ordeals have also been well-studied in economics, both as a method to elicit information without using monetary payments, and in the efficiency-equity trade-off when using in-kind transfers [Nichols and Zeckhauser, 1982]. However, utility maximization and ordeals have minimal prior work from the algorithmic mechanism design perspective.

As mentioned earlier, Hartline and Roughgarden completely resolved the question of optimal utility maximization in the single-dimensional Bayesian setting. An earlier paper by [Chakravarty and Kaplan, 2006] also studies when, for the single-dimensional setting of multi-unit unit-demand, the utility-optimal mechanism gives randomly gives items away for free. Condorelli [2013] produces a similar theory by extending Myerson [1981] to characterize optimal mechanisms for settings both with and without money, and parameterizes these results in an agent's willingness-to-pay, which does not have to be equal to an agent's value.

Recent work has studied variations on ordeals and money burning, but most of these settings are single-dimensional. Motivated by health insurance, Essaidi et al. [2024] investigate whether, in equilibrium, limiting entry of sellers into a two-sided market improves consumer surplus or not. Dworczak et al. [2023] study a different but related problem, the allocation of (divisible) money, and characterize when using payments via ordeals to better allocate the money yields more utility than ordeal-free equal payments to all agents. Patel and Urgun [2022] consider the interplay between verification, costly only to the designer, and ordeals, costly to both, in the single-dimensional i.i.d. setting. Lundy et al. [2019] also study the power of verification, but in the setting of allocating indivisible resources to food banks using wait times as an ordeal. Lundy et al. [2024] study utility maximization in the context of mobile games that care about platform retention.

A few recent works foray into multidimensional settings. Ashlagi et al. [2021] study the objective function that is a weighted sum of allocative efficiency and social welfare, which can include utility, for the setting of unit-demand agents for totally-ordered objects, restricted to i.i.d. This setting was introduced in [Fiat et al., 2016] and proven to be in between single- and multi-dimensional environments by several complexity metrics. Braverman et al. [2016] study a truly multidimensional problem, the allocation of heterogeneous healthcare services to unit-demand patients where wait times serve as ordeals, but they focus on the computation of efficient equilibria. Yang [2022] and Akbarpour et al. [2023] study a multidimensional ordeal setting that is quite different from what we study. The mechanism designer can allocate either a good (which has positive value) or an ordeal (which has negative value) and can also use monetary payments (which are transferable and do not impact the objective function). They derive conditions under which it is optimal to only allocate the good and not an ordeal, and under which one ordeal mechanism dominates another.

Note that while there is ample work on mechanism design *without* money (e.g. [Psomas and Verma, 2022, Fotakis et al., 2014, Goko et al., 2022]), it is not relevant to this work, as we are particularly exploring the setting with payments that hurt the objective, and the trade-off of using them.

## A.1    Relationship to Subsequent Work

As mentioned in Section 1.1, very recent subsequent work by Ezra et al. [2025] builds on our work in the i.i.d. setting to also designs mechanisms whose utility approximates the benchmark of optimal social welfare. Most salient, they achieve a general $O(1 + \log \frac{cn}{m})$-approximation for the condition that the probability of any item being an agent's favorite is at most $c/m$, characterizing the improvement of utility in $m$ and $c$. This implies a $\Theta(\log n)$-gap between optimal utility and social welfare for general unit-demand bidders and matches our $O(\log \frac{n}{m})$-approximation for the i.i.d. unit-demand setting (and in fact, for the uniform favorites setting of Appendix C).

They consider two additional settings—multi-unit auctions where buyers have submodular valuations, and auctions with divisible goods where buyers have concave valuations—in both of which they match this $\Theta(\log n)$-gap.

In contrast, our paper is the *first* to study classical (unit-demand) multidimensional Bayesian utility maximization. As part of this, we provide a lengthy related work (Appendix A) and discussion on directions and approaches (Appendix D). Unlike Ezra et al. [2025], we provide matching lower bounds (Section 3.4). Finally, we highlight that a main contribution of this work is the simplicity of both our mechanisms and analysis, which is the result of substantial effort and refinement of complex arguments to highlight their intuition. We compare the simplicity and practicality of our mechanism to that of Ezra et al. [2025] at the end of Section 3.1.

An additional subsequent work by Ganesh and Hartline [2025] studies the problem of combinatorial pen testing using deferred acceptance mechanisms, which is equivalent to a Bayesian utility maximization setting with single-dimensional agents and service feasibility constraints. They achieve a similar gap between optimal utility and optimal welfare of $(1 + o(1)) \cdot \ln(n)$ for $n$ pens (buyers). However, their mechanism design approach of reducing to deferred auctions relies virtual values and quantile-based transformations, and is therefore more complex to ours. Further, their work is only in the single-dimensional setting.

**Simple-Yet-Approximately-Optimal Mechanisms for Seller Revenue.** While single-dimensional Bayesian revenue maximization is solved in closed-form by Myerson's Nobel-prize-winning theory [Myerson, 1981], in the multidimensional setting, even selling two heterogeneous goods to a single additive buyer can be intractable, requiring infinitely many randomized options to extract optimal revenue [Manelli and Vincent, 2006, Daskalakis et al., 2017]. As a result of this intractability, a large body of work has instead focused on approximation. In particular, this work has designed *simple* mechanisms that yield constant-factor approximations to the optimal revenue in a variety of settings [Cai et al., 2016, Chawla et al., 2010, Babaioff et al., 2014, Cai and Zhao, 2017, Eden et al., 2017, Cai et al., 2019, Yao, 2015, Chawla and Miller, 2016, Rubinstein and Weinberg, 2015], developing a barrage of techniques and insights along the way. There are two main steps required in these approaches: finding an upper bound on the optimal revenue, and then determining which simple mechanisms can be used to approximate this benchmark. For most of these settings, the mechanisms are some variation on either selling all of the items separately, or selling them all together in one grand bundle.

## B   Proof of Lemma 3

In this appendix we present a proof of the lemma we use in our analysis in Section 3.3. If the number of bidders who favor an item was independent from the maximum favorite-value for that item (that is, bidder values for that item such that it is their favorite), this lemma would be immediate. However, because these are correlated, we instead require this lemma. It is a somewhat technical application of various probability tools, and relies on the fact that the maximum of bidder favorite-values for an item, when values are drawn i.i.d., is distributed nicely.

*Proof.* Using that $n'$ is drawn according to $\mathrm{Binomial}(n, 1/m)$, we want to show that the left-hand side is at least a constant times $\mathbb{E}[V]/(1 + \log n/m)$. We do this by breaking the left-hand side into two events: when $n'$ is "close" to its expectation $n/m$ (say, at most 10 times its expectation), and when it is not. Then we can write

$$\mathbb{E}\left[\frac{V}{1 + \log n'} \mid V \geq v'\right] \geq \mathbb{E}\left[\frac{V}{1 + \log n'} \mid V \geq v' \cap n' \leq 10\frac{n}{m}\right] \Pr[n' \leq 10\frac{n}{m} \mid V \geq v']$$

$$\geq \frac{\mathbb{E}[V \mid V \geq v' \cap n' \leq 10n/m] \cdot \Pr[n' \leq 10n/m \mid V \geq v']}{4.32(1 + \log n/m)}.$$

Because $n' \leq 10n/m$, then $1 + \log n' \leq 1 + \log 10n/m \leq (1 + \log 10)(1 + \log n/m)$ for $n/m \geq 1$, and $1 + \log_2 10 \approx 4.32$. As a result, we only need to compare the numerators. We now work on the right-hand side's numerator of $\mathbb{E}[V]$ to get it into a comparable state:

$$\mathbb{E}[V] \leq \mathbb{E}[V \mid V \geq v'] = \mathbb{E}[V \mid n' \leq 10n/m \cap V \geq v'] \cdot \Pr[n' \leq 10n/m \mid V \geq v']$$

$$+ \mathbb{E}[V \mid n' > 10n/m \cap V \geq v'] \cdot \Pr[n' > 10n/m \mid V \geq v']$$

Further, we see that and $E[V \mid n' > 10n/m \cap V \geq v'] \leq 2\mathbb{E}[V \mid n' > 10n/m]$:

$$\mathbb{E}[V \mid n' > 10n/m] = \mathbb{E}[V \mid n' > 10n/m \cap V \geq v'] \cdot \Pr[V \geq v' \mid n' > 10n/m]$$
$$+ \mathbb{E}[V \mid n' > 10n/m \cap V \leq v'] \cdot \Pr[V < v' \mid n' > 10n/m]$$
$$\geq \mathbb{E}[V \mid n' > 10n/m \cap V \geq v'] \cdot \frac{1}{2}$$

where this follows from the fact that $1/2 \leq \Pr[V \geq v'] \leq \Pr[V \geq v' \mid n > 10n/m]$ and from ignoring the second term.

And furthermore,

$$\mathbb{E}[V \mid n' > 10n/m] \leq \mathbb{E}_{n'}[\mathbb{E}_X[X \cdot n' \mid n' > 10n/m]] \qquad \max_{n'} X \leq \sum_{n'} X$$
$$\leq \mathbb{E}[X] \cdot \mathbb{E}[n' \mid n' > 10n/m]$$
$$\leq \mathbb{E}[X] \cdot (10n/m + n/m). \qquad\qquad n' \text{ Binomial.}$$

The first inequality is due to the max of n' draws being at most the sum of $n'$ draws. The third inequality follows from the fact that a binomial distribution conditioned on some number of successes and increased by the same number of trials is memoryless, so just conditioned on conditioned on some number of successes and with no increase in the number of trials, the expectation is less than as if it were memoryless. Therefore,

$$\mathbb{E}[V \mid n' \geq 10n/m \cap V \geq v'] \leq 2 \cdot \mathbb{E}[X] \cdot (11n/m).$$

Together, this gives

$$\mathbb{E}[V] \leq \mathbb{E}[V \mid n' \leq 10n/m \cap V \geq v'] \cdot \Pr[n' \leq 10n/m \mid V \geq v']$$
$$+ 2 \cdot \mathbb{E}[X] \cdot (11n/m) \cdot \Pr[n' > 10n/m \mid V \geq v']$$

The first term appears on the left-hand side. To get a constant-approximation, we aim to bound the probability $\Pr[n' \geq 10\frac{n}{m} \mid V \geq v']$ to make the second term negligible. We will first use a Chernoff bound on $\Pr[n' \geq 10\frac{n}{m}]$. The Chernoff bound gives

$$\Pr[n' \geq (1 + \delta)\frac{n}{m}] \leq exp(-\frac{\frac{n}{m}\delta^2}{2 + \delta}).$$

Then for $\delta = 9n/m$, we get

$$\Pr[n' \geq 10\frac{n}{m}] \leq exp(-\frac{\frac{n}{m}9^2}{11}) < exp(-7.36n/m).$$

Then using Bayes' rule,

$$\Pr[n' \geq 10\frac{n}{m} \mid V \geq v'] = \frac{\Pr[n' \geq 10\frac{n}{m}] \cdot \Pr[V \geq v' \mid n' \geq 10\frac{n}{m}]}{\Pr[V \geq v']} \qquad (*)$$
$$\leq 2\Pr[n' \geq (1 + \delta)\frac{n}{m}])$$
$$= 2exp(-7.36n/m)$$

where $(*)$ is because $\Pr[V \geq v'] \geq 1/2$ and of course, $\Pr[V \geq v' \mid n' \geq 10\frac{n}{m}] \leq 1$.

All together, since $\mathbb{E}[X] \leq \mathbb{E}[V]$ by definition,

$$\mathbb{E}[V] \leq \mathbb{E}[V \mid n' \leq 10n/m \cap V \geq v'] \cdot \Pr[n' \leq 10n/m \mid V \geq v']$$
$$+ 2 \cdot \mathbb{E}[X] \cdot 22\frac{n}{m} \cdot e^{-7.36n/m}$$
$$(1 - 44\frac{n}{m} \cdot e^{-7.36n/m})\mathbb{E}[V] \leq \mathbb{E}[V \mid n' \leq 10n/m \cap V \geq v'] \cdot \Pr[n' \leq 10n/m \mid V \geq v'].$$

And for all $\frac{n}{m} \geq 1$, $44\frac{n}{m} \cdot e^{-7.36n/m}$ is upper-bounded by some small constant $c < 1$, precisely, 0.028, hence $4.32/(1 - 0.028) < 4.45$. $\qquad\qquad\square$

## C  Complications Beyond i.i.d.

We now reexamine the two places where we use the i.i.d. assumption in our main results. First, the incentive-compatibility guarantees of the $\mathcal{M}$-Favorites mechanism proven in Lemma 1 relied highly on symmetry. Second, some of our approximation analysis does as well, but we suspect known techniques may extend it beyond i.i.d.

One easy extension would be to move beyond i.i.d. across both bidders and items to only i.i.d. across bidders, matching the standard assumptions in the multidimensional prior-independent literature. We look at where we have used assumption that item values are drawn i.i.d. At first glance, it seems to be only leveraging the fact that each of single-item subroutines is identical and has the same number of participants in expectation. A natural weaker condition would be to just assert this directly—that is, a setting in which, for each bidder, the probability that any given item is their favorite is uniform across items. We call this setting the *uniform favorites* setting.

While the Random-Favorites mechanism used in Theorem 1 remains BIC in the uniform favorites setting—an agent's utility depends only on one's value and the expected number of competitors—the Prior-Free-Favorites mechanism used in Theorem 2 does not.

**Theorem 5.** *In the uniform favorites setting, the Prior-Free-Favorites mechanism is not guaranteed to be BIC.*

Recall that the single-item prior-free mechanism is detailed in Definition 2. Informally, a random price is chosen doing the following: bidders are ordered by report, a power-of-two is chosen uniformly at random, and then value $v_{2^j+1}$ is used as a price (where $v_{n+1} = 0$). The mechanism allocates to a bidder chosen uniformly at random from those who reported above this price.

*Proof.* We construct an example for the uniform favorites setting that violates BIC. Consider $n$ bidders and 2 items, where each bidder draws their values i.i.d. from the following item distributions: $v_{i1} \sim U(0,1)$ and $v_{i2} \sim U\{0,1\}$. This is a uniform favorites setting, as each buyer prefers each item with probability $0.5$.

Consider an agent $\ell$ with values $v_\ell = (1-\epsilon, 1)$. If they report their true preference of item 2, then they are placed in contention for it. However, by our construction, agents prefer item 2 if and only if they have a value of 1. Hence, in order to earn non-zero utility from item 2, the random price of the single-item prior-free subroutine used in Prior-Free-Favorites must not be a competitor's price; it must choose the price $v_{n'+1} = 0$, which occurs with probability $\frac{1}{1+log(n')}$. In this case, agent $\ell$ wins the item uniformly at random with probability $1/n'$, hence their expected utility is $\frac{1}{1+log(n')}(\frac{1}{n'})$.

On the other hand, if the agent misreports $v'_\ell = (1-\epsilon, 0)$, they are placed in contention for item 1. Then for the random price $v_{n'+1} = 0$, the agent gets utility $(1-\epsilon)\frac{1}{n'}$, and in the event of any other random price, they gain some strictly positive expected utility that depends only on $n'$. Therefore, there exists $\epsilon > 0$ such that this misreport provides strictly higher utility than truthful reporting. $\square$

As even this slight weakening of i.i.d. can cause a violation of BIC for $\mathcal{M}$-favorites depending on $\mathcal{M}$, our results are restricted to the setting where values are i.i.d. across both bidders and items.

**Remark 4** (On only i.i.d. items). While Lemma 1 only requires i.i.d. across items, our analysis breaks down without i.i.d. bidders as well, as Lemma 2 is no longer necessarily true. One bidder's distribution for all items may be far higher than all others, and thus the probability of favorites aligning could be minuscule.

## D  Discussion: Complexities of General Unit-Demand

Building on Section 4, In this section, we discuss the complexities of the general unit-demand setting beyond when values are drawn i.i.d. We touch on the structure of multidimensional utility-optimal mechanisms and how they leverage the power *not* to allocate.

## D.1 Optimal Multidimensional Mechanisms

We present a surprising lack of structure on the *utility-optimal* mechanism for the general unit-demand setting, which in turn motivates the study of simplicity and approximation. We begin by reviewing what is known from the single-dimensional setting.

**Single-dimensional Mechanisms.** In the single-parameter setting, the optimal mechanism *always* allocates the items, it's just a question of to who and for how much. The mechanism takes an analogous structure to the revenue-optimal mechanism, maximizing "ironed" *virtual* welfare, where for utility maximization, a buyer's virtual value is $\theta_i(v_i) = \frac{1-F_i(v_i)}{f_i(v_i)}$. To iron into $\bar{\theta}_i$, this function is averaged until it is monotone non-decreasing in $v_i$. Then, if there is one item, give it to the bidder with the highest ironed virtual value $\bar{\theta}_i(v_i)$.

What this mechanism concretely looks like depends highly on the *hazard rate* of the buyer's type distribution, $\frac{f_i(\cdot)}{1-F_i(\cdot)}$. In one extreme case, when the hazard rate is non-decreasing (monotone hazard rate, or MHR), $\bar{\theta}_i$ is fully averaged, resulting in a mechanism that allocates uniformly at random and charges no payments. In the other extreme, when the hazard rate is non-increasing (anti-MHR), $\theta_i$ does not need to be averaged as it is already monotone non-decreasing, and the mechanism looks like a second-price auction but in this virtual space.

**Multidimensional Mechanisms.** From this characterization, we can already expect the optimal multidimensional mechanism to have cases that make it look like VCG, requiring payments, and cases that make it look like random allocations with no payments. However, we conclude that getting a handle on the structure of this mechanism may be highly difficult.

Despite the goal being to maximize utility—a dual goal of giving away all of the items most efficiently and charging as minimal payments as possible—it turns out we can't even rule out the possibility that the optimal mechanism *does not allocate* an item in order to better allocate the other items and avoid using payments. Note that this contrasts with the single-dimensional setting, where the optimal mechanism *does* always allocate! Mathematically, this means that we cannot represent the optimal allocation as a convex combination of deterministic exhaustive allocations.

**Theorem 6.** *If we are maximizing utility over all allocations and payments, it is* not *without loss to restrict attention to convex combinations of deterministic exhaustive allocations:* $\sum_i x_{ij}(\mathbf{v}) = 1 \, \forall j, \mathbf{v}$. *There are instances where it maximizes utility not to allocate an item $j$ in every instance.*

Note that this applies to the general Bayesian utility maximization problem, and is not restricted to the i.i.d. setting—our construction will not be i.i.d.

*Proof.* We construct a setting in which the optimal mechanism must sometimes discard an item. Consider a setting with two unit-demand bidders and two items. Bidder 1 has values $(v_{11}, v_{12}) = (1, 3)$ deterministically. For some $c$, bidder 2 has two potential type profiles with equal probability,

$$(v_{21}, v_{22}) = \begin{cases} (c, c+1) & \text{w.p. } 1/2 \\ (1, 4) & \text{w.p. } 1/2. \end{cases}$$

The efficient allocation is $x_2(1, 4) = (0, 1)$ and $x_2(c, c+1) = (1, 0)$, allocating the other item to bidder 1, yielding expected welfare $0.5c + 4$. However, we must charge a payment of 1 to the type $(1, 4)$ to prevent the bidder with type $(c, c+1)$ from misreporting, and this hurts utility. We now show that any convex combination of deterministic allocations loses an additive $\frac{1}{2}$ utility compared to welfare.

Assume our mechanism *is* a convex combination of deterministic allocations. First observe that because we have an equal number of bidders and items, for each type profile, not only do the allocation probabilities sum to 1 for each item across bidders, but they also must sum to 1 for each *bidder* across items. As $v_1$ is fixed, we'll write our input to $x_2(\cdot)$ and $p_2(\cdot)$ as $v_2$ rather than $\mathbf{v}$. Now, for some probabilities $0 \leq \alpha, \beta \leq 1$, let $x_2(c, c+1) = (\alpha, 1 - \alpha)$ and $x_2(1, 4) = (\beta, 1 - \beta)$ (where bidder 1 is allocated whatever bidder 2 is not by assumption). Then the resulting welfare from these

allocations is

$$\text{WELFARE} = 0.5 \left( \underbrace{\alpha c + (1-\alpha)(c+1)}_{\text{bidder 2}} + \underbrace{(1-\alpha) + 3\alpha}_{\text{bidder 1}} \right) + 0.5 \left( \underbrace{\beta + 4(1-\beta)}_{\text{bidder 2}} + \underbrace{(1-\beta) + 3\beta}_{\text{bidder 1}} \right)$$
$$= 0.5(c + 2 + \alpha) + 0.5(5 - \beta).$$

The payment we must charge bidder 2 to prevent misreporting is $p_2(1,4) = \alpha - \beta$, giving utility

$$\text{UTILITY} = 0.5(c + 2 + \alpha) + 0.5(5 - \alpha)$$
$$= 0.5c + 3.5.$$

Alternatively, we could offer the allocation $x_2(1,4) = (0, \frac{c}{c+1})$. Notice that the type $(c, c+1)$ is indifferent between this allocation and the allocation $x = (1, 0)$, and therefore requires no payment for incentive-compatibility. This allocation achieves the efficient outcome with probability $\frac{c}{c+1}$ and therefore is arbitrarily close to welfare as $c$ grows large. $\qquad \square$

The construction in this proof actually highlights important intuition about preventing misreporting via payments versus allocations.

**Observation 1.** If the additive gap between an agent's favorite type and allocated type is small, it is best for utility to maintain incentive-compatibility with small payments, i.e., money burning. In contrast, if the ratio between these types is small (as in the proof), we can best maintain incentive-compatibility with a fractional allocation, i.e., very rarely "burning" a good.

This result evidences complexity in optimal mechanisms, and combined with the environment's similarities to revenue maximization, provides a convincing argument that designing utility-optimal mechanisms is extremely likely to be intractable. As a result, we focus on approximation.

