# OpenReview forum: "Multidimensional Bayesian Utility Maximization: Tight Approximations to Welfare"
_NeurIPS.cc/2025/Conference — NeurIPS 2025 spotlight_

### Official Review · Reviewer_ofs2 · 2025-06-29

**Clarity:** 4
**Significance:** 2
**Originality:** 2
**Rating:** 4
**Confidence:** 4

**Summary:**

This paper studies utility maximization mechanism design in the multidimensional setting. Specifically, they consider the specific sub-case of unit-demand buyers, whose value towards each item is drawn i.i.d. from the same distribution $\mathcal{D}$. Their benchmark is against the maximum welfare of the instance. It is worth noting that this is different (albeit similar) to a previous model by [Hartline, Roughgarden '2008], in which each buyer has a single value for all the items.

They developed a simple and prior-independent class of mechanism, called "Favorites" mechanism, that first bucket the buyers by their favorite items, then run a prior-free mechanism $\mathcal{M}$ per each item. They showed that:
- For the case where there are more items than buyers, the mechanism $\mathcal{M}$ that randomly allocates the item to a buyer that has it as their favorite guarantees a $1 - 1/e$ approximation.
- For the case where there are more buyers than items, applying the mechanism introduced in [Hartline, Roughgarden '2008] as $\mathcal{M}$ guarantees a $O(\log n/m)$ approximation.

Furthermore, they demonstrate that these approximations are tight, due to a lower bound of $\Omega(1 + \log n/m)$ derived similarly to that in [Hartline, Roughgarden '2008].

**Questions:**

**Questions**

1. (Related to weakness 2 above) I believe there is a direct reduction from the i.i.d. model in this paper to the identical-item model in the [Hartline, Roughgarden '2008] paper as follows: for each buyer, replace all item values with the highest item value for this buyer. In fact, this is exactly what Definition 3 is doing for the tightness result, and (I believe) also exactly what the Favorites mechanism class is doing: it is ignoring all non-favorite items for all buyers. Can you provide evidence that this is (or is not) the case?
2. What can you do against a different benchmark, for example the optimal utility achievable? I find comparing utility to welfare is a bit like comparing rotten apples to apples, but I understand that this has been the common benchmark for utility maximizing results.

**Comments for authors**:
1. It is minor, but in the proof for the $1-1/e$ mechanism, you should have stated that the probability that each agent is allocated is $1-1/e$ **independent** of how their item values are drawn (otherwise, you can imagine an argument that says that they have higher probability of being allocated if their value for their favorite item is smaller, which is not the case here).

**Ethical Concerns:**

["NO or VERY MINOR ethics concerns only"]

**Final Justification:**

I want to update my score from 3 to 4, as I found the authors have given enough justifications for the existence of this paper in the rebuttal, despite superceding results by [Ezra et. al' 2025]. This was the main reason for my original verdict.

**Limitations:**

yes

**Quality:**

4

**Strengths And Weaknesses:**

**Strengths**:
- The paper is well-written, and the relationship of the set of results of the paper against other related results are properly discussed (especially to those of [Hartline, Roughgarden '2008] and [Ezra et. al '2025]).
- The mechanisms introduced are very simple and easy to implement yet optimal, and I believe that would be incredibly practical if being used in applications. The proofs of the technical results are also well-written and clear.

**Weakness**:
- This set of results is superseded by that of [Ezra et. al '2025], who considered the case of unit-demand non-identical buyers, and the values of the items for each buyer can be correlated. The other paper also recovers the results of this paper (and beyond), albeit with a more complicated mechanism. In that sense, I believe the main contribution of this paper is proving the existence of **simple** mechanisms that gets the job done for the i.i.d. case.
- This paper also relies heavily on the previous result of [Hartline, Roughgarden '2008] (especially tightness result and the $n > m$ case), to the extent that I believe the i.i.d. model of this paper can simply be reduced to the identical-item model of [Hartline, Roughgarden '2008].

---

> ### Author Rebuttal · Authors · 2025-07-28
>
> > "This set of results is superseded by that of [Ezra et. al '2025], who considered the case of unit-demand non-identical buyers, and the values of the items for each buyer can be correlated. The other paper also recovers the results of this paper (and beyond), albeit with a more complicated mechanism. In that sense, I believe the main contribution of this paper is proving the existence of simple mechanisms that gets the job done for the i.i.d. case."
>
> It is true that the follow-up work by [Ezra et. al '2025] is able to (1) match our result in our setting and (2) get a slightly weaker result in the more general setting; however, we do not necessarily view this as a weakness. First, we think our paper makes its own contributions. As you point out, we achieve the same results with a simpler mechanism, which as you also point out, is important for our motivating applications. In addition, we are  first paper in the space of sublinear approximations for multidimensional utility maximization, and so we include a collection of smaller results and discussions that contextualize the difficulty of the problem as a whole. Finally, we think the existence of follow-up work which builds upon our work shows the importance of our paper and this direction itself for the field.
>
> > "(Related to weakness 2 above) I believe there is a direct reduction from the i.i.d. model in this paper to the identical-item model in the [Hartline, Roughgarden '2008] paper as follows: for each buyer, replace all item values with the highest item value for this buyer. In fact, this is exactly what Definition 3 is doing for the tightness result, and (I believe) also exactly what the Favorites mechanism class is doing: it is ignoring all non-favorite items for all buyers. Can you provide evidence that this is (or is not) the case?"
>
> This is a great question, and in our opinion, the right intuition to have about why the favorite mechanism works in the first place. However, such a reduction is not so straightforward when trying to prove a positive result (i.e. that this approximation ratio can be achieved by some implementable mechanism in our setting). Namely, performing a reduction where "for each buyer, replace all item values with the highest item value for this buyer" would require knowing a bidder's private information. Therefore, to implement such a reduction requires that bidders are incentivized to report this information truthfully given knowledge of the mechanism. We prove that this is the case in the first step of our proof (and interestingly would not be true under more stringent DSIC constraints). We would then need to show that ignoring the the rest of the information about bidders' types does not harm welfare by too much---this would require an analysis that is equivalent to much of the technical steps in our proof. So while reframing the proof as a reduction might work, we do not believe it would simplify the analysis, nor would it immediately give a mechanism that could be used to achieve the approximation which is an added bonus of our constructive proof.
>
> > "What can you do against a different benchmark, for example the optimal utility achievable? I find comparing utility to welfare is a bit like comparing rotten apples to apples, but I understand that this has been the common benchmark for utility maximizing results."
>
> This is also a good question and we agree that a comparison to optimal utility would be more satisfying (it's the focus of much of our ongoing work). However, this turns out to be a much more difficult problem (we give a discussion as to why it is difficult in Appendix D.2) which is why welfare is currently used as a benchmark in utility maximization. We do agree though that finding tighter benchmarks, whether they be optimal utility or something in between optimal utility and welfare, is exciting future work. \textbf{though likely a herculean task? or say something about how much more difficult this is going to be?}
>
> > "Comments for authors: It is minor, but in the proof for the mechanism, you should have stated that the probability that each agent is allocated is independent of how their item values are drawn (otherwise, you can imagine an argument that says that they have higher probability of being allocated if their value for their favorite item is smaller, which is not the case here)."
>
> Agreed, thanks for pointing this out.

---

> > ### Comment · Reviewer_ofs2 · 2025-08-02
> >
> > Thank you for your thorough response! I also agree that comparing utility to utility is a very difficult task in general, and pulling off such feats would be very satisfying.

---

### Official Review · Reviewer_NShc · 2025-07-02

**Clarity:** 3
**Significance:** 3
**Originality:** 3
**Rating:** 5
**Confidence:** 4

**Summary:**

This paper studies the problem of Bayesian utility maximization in the setting with n unit-demand agents and m items, where the valuation  of an agents for an item is drawn independently and identically (so that an agent could have different valuations for different items). This generalize the classic results with n unit-demand agents and m identical items (where an agent has the same valuation across items).

This paper proposes two simple mechanisms with provable approximation guarantees, Random-Favorites for the case with m >= n and Prior-Free-Favorites in the n > m case. Both mechanisms ask the agents to report their favorite items first. Random-Favorites simply allocates the item to a random agent who indicates the item as favorite while Prior-Free-Favorites applies the mechanism from the setting with n unit-demand agents and m identical items.

The authors further demonstrate tightness of their results.

**Questions:**

Minor comment:

1. It is unclear whether this paper is using an approximation ratio that is smaller than 1 (i.e., A smaller number indicates a worse approximation) or larger than 1 (i.e., A larger number indicates a worse approximation). In particular, (1 - 1/e) is a number smaller than 1. But log(n/m) when n > m can go from a number smaller than 1 to a number larger than 1, while when n gets reasonably large, log(n/m) is larger than 1.

**Ethical Concerns:**

["NO or VERY MINOR ethics concerns only"]

**Final Justification:**

This paper initiates a new research direction, provides elegant results, and has already inspired follow-up papers. I will raise my score by 1.

**Paper Formatting Concerns:**

None.

**Quality:**

3

**Strengths And Weaknesses:**

Strength:

1. The paper is generally well-written, clear, and easy to follow. The problem studied in this paper is theoretically interesting, and of interest in the AGT community.
2. This paper initiates a new research direction and the collection of results it provided is technically non-trivial.

Weakness:

1. (This might not be a weakness) This paper is not a standard submission since there is a follow-up paper with stronger results (which cover all results in this paper as special cases) in STOC'25.

---

> ### Author Rebuttal · Authors · 2025-07-28
>
> > "(This might not be a weakness) This paper is not a standard submission since there is a follow-up paper with stronger results (which cover all results in this paper as special cases) in STOC'25."
>
> Thank you for pointing this out---we believe the fact that our paper has already generated follow-up work to be a strength.
>
> > "It is unclear whether this paper is using an approximation ratio that is smaller than 1[...]"
>
> You're correct that we often discuss approximations as larger than 1 when we mean less than 1.  We will clarify this in our final version.

---

> > ### Comment · Reviewer_NShc · 2025-08-01
> >
> > Thank you for the response!

---

### Official Review · Reviewer_igYG · 2025-07-03

**Clarity:** 4
**Significance:** 2
**Originality:** 3
**Rating:** 5
**Confidence:** 2

**Summary:**

In this paper, the authors consider the problem of Bayesian utility maximization for $n$ unit-demand buyers and $m$ heterogeneous items. The goal of the designer is to maximize the utility (which is welfare minus payments). This setting generalize the prior work of Hartline and Roughgarten(2008) who considered the case of $m$ identical items. The authors propose a simple scheme called M-favorites and characterize the bound on the ratio of utility over welfare when $m \geq n$ (many items regime) and $n > m$ (many bidders regime).

**Questions:**

**Q1:** I wonder if the i.i.d. assumption across items and bidders necessary for proving Lemma 1? Can we relax it to something more general, like exchangeability across rows and identical across bidders?

**Q2:** Can you discuss the computational cost of implementing the scheme in the $m \geq n$ and $n>m$ cases?



Minor Questions / Comments

* Line 168: "it is without loss to restrict attention to allocation rules that allocate at most one item to each agent, that is to have feasibility constraint $\sum_{j} x_{ij}(v) \leq 1$". But isn't this feasibility constraint in expectation? Because in the previous paragraph it is mentioned that $x_{ij}(v)$ is the probability that bidder $i$ receives item $j$.

* Line 179: The beginning of the sentence seems to be missing?

**Ethical Concerns:**

["NO or VERY MINOR ethics concerns only"]

**Final Justification:**

I enjoyed reading this paper, and I think this is an interesting contribution (from the perspective of an external researcher to this area). I also have a better understanding of the paper by reading the authors' response to reviewer ofs2. Based on that, I think I will retain my previous score.

**Limitations:**

Yes.

**Quality:**

3

**Strengths And Weaknesses:**

Strengths

* I think the paper is really well written. I do not work in this area of research, and still I was able to follow the construction and the results.

* The M-favorites scheme seems quite elegant as it reduces the multidimensional problem to several single-item problems that are addressed via either random or the prior-free mechanism of Hartline-Roughgarden.

Weaknesses

* I did not observe any obvious weaknesses, other than the fact that some of the assumptions seem quite strong (i.i.d. valuations across items and bidders, unit demand etc.). It would be helpful if the authors include some discussion on possible relaxation of these assumptions in the main text.

---

> ### Author Rebuttal · Authors · 2025-07-28
>
> You raise a good point that assuming i.i.d. across both bidders and items is quite strong. We discuss the potential (or lack thereof) for relaxing the bidder or item i.i.d. constraints in Appendix C, "Complications Beyond i.i.d.," which was moved to the appendix due to space constraints, but perhaps we could move some high level intuition back into the main text. We also know that in the general case where items are not i.i.d. one cannot achieve better than $log(n)$, from the simple case where every bidder greatly prefers the same item reducing it the the single item case from Hartline and Roughgarden (2008).
>
> Line 168: Yes, the feasibility is in expectation over the allocation of items, but it is still without loss to impose the constraint because it would never benefit the mechanism to allocate beyond it.  You're correct that the strictest constraint we could impose would require correlation or deterministic allocations.
>
> Line 179: Thanks for pointing this out: the end of the equation should be a comma and no new line, and the partial sentence refers to the equation.

---

> > ### Comment · Reviewer_igYG · 2025-08-05
> > **Response to the authors**
> >
> > Thank you for your answers to my questions, and in particular for pointing me to the discussion on the non i.i.d. case. My impression of this paper remains positive, and I will retain my previous score.

---

### Official Review · Reviewer_8LRZ · 2025-07-03

**Clarity:** 4
**Significance:** 3
**Originality:** 2
**Rating:** 5
**Confidence:** 3

**Summary:**

The paper considers multidimensional Bayesian utility maximization, with an emphasis on a unit-demand environment where values are independently and identically distributed (i.i.d.) over items and buyers alike. As such, it seeks to extend the foundational work of Hartline and Roughgarden (2008) from a single-dimensional environment to a multidimensional one. The authors propose simple, prior-independent mechanisms that yield approximations to optimal social welfare.

The paper contains two main results: For the case with more goods than agents (m≥n), their mechanism achieves a $(1−1/e)$-approximation of utility to maximum welfare.

For the cases with more agents than goods $(n>m)$, it achieves an $O(\log n / m)$-approximation.

The authors establish these bounds are tight in both n and m, which leads to an overall $\Theta(1+log n/m)$ gap between optimal utility and social welfare.

To achieve these guarantees, they introduce "Favorites Mechanisms," where each buyer indicates their favorite good, and allocation progresses item-by-item utilizing specific single-item mechanisms for each item-buyer ratio. The paper also includes a precise description of the difficulties of Bayesian utility maximization in multiple dimensions, explaining why known techniques in revenue maximization literature are not directly applicable. One counter-intuitive finding is that an optimal utility mechanism in some cases has to leave a good intentionally unallocated in order to be optimal.

**Questions:**

The paper highlights that effective utility mechanisms can "throw away" objects. Could the authors give some more discussion of the real-world implications of this finding in social service allocation problems, possibly in the form of further concrete examples beyond the thought experiment argument?

Aside from theoretical lower bounds, are there specific empirical conditions or problem structures under which the current mechanisms might hit performance limits, even in the i.i.d. case?

**Ethical Concerns:**

["NO or VERY MINOR ethics concerns only"]

**Final Justification:**

Given the detailed response by the authors, I'm inclined to increase my score by 1.

**Limitations:**

Yes

**Quality:**

3

**Strengths And Weaknesses:**

Strengths

The authors obtain tight and near-optimal approximation guarantees for utility relative to social welfare, taking a first step to generalising a classic result in single-dimensional problems to the more complicated multidimensional case.

Innovative Problem Formulation: The paper formulates a serious research on Bayesian maximization of multidimensional utility, which is a less mature field than revenue or welfare maximization. It highlights the relevance in social service allocation where non-monetary "payments" (ordeals) are provided.

The "Favorites Mechanisms" proposed are prior-independent (i.e., do not require complete knowledge of the distributions beneath), which makes them more practical.

I think the paper is well written, the main contributions are presented in a clear manner.

Weaknesses

The key results are heavily reliant upon the assumption that item values are i.i.d across items as well as buyers. The authors note that relaxing this assumption, e.g., to non-i.i.d. bidders, can make basic properties like incentive compatibility or the analysis invalid.

Despite the general simplicity of the overall architecture, the "Prior-Free-Favorites" mechanism for the $n>m$ case draws on the one-dimensional prior-free mechanism of Hartline and Roughgarden (2008) as a subroutine. The implementation details in practice and computational expense of these subroutines are not fully detailed.

---

> ### Author Rebuttal · Authors · 2025-07-28
>
> Thank you for the thoughtful review, we respond to your points below.
>
> > "Despite the general simplicity of the overall architecture, the "Prior-Free-Favorites" mechanism for the
>  case draws on the one-dimensional prior-free mechanism of Hartline and Roughgarden (2008) as a subroutine. The implementation details in practice and computational expense of these subroutines are not fully detailed."
>
> The implementation details of the subroutine mechanism are specified in Definitions 2. Thank you for pointing out that we do not specify the running time.  The running of the Favorites mechanisms is $O(n+mR)$ where $R$ is the running time of the single-item mechanism.  For the lottery, $R$ is $O(1)$.  For the Harline Roughgarden subroutine, $R$ is $O(n')$ where $n'$ is the number of bidders participating in this mechanism.
>
> > Re: throwing away objects:
>
> In real world allocations like social services, this more corresponds to services going (temporarily) unallocated, such as a shipment of food that is not allocated to any food bank or an affordable housing unit being left vacant. Our result says: by not fully allocating these services and good, the mechanism can subsidize the costs to those who are allocated and charge them less (in ordeals such as bureaucracy or wait times).
>
> We acknowledge that such a solution might be unpalatable in the real world (indeed Lundy et. al discuss this fact in their setting on allocating food to food banks). It is an interesting open question to ask how much efficiency would be lost by restricting to the class of mechanisms which fully allocate.
>
> > "Aside from theoretical lower bounds, are there specific empirical conditions or problem structures under which the current mechanisms might hit performance limits, even in the i.i.d. case?"
>
> Interesting question, in the case of $m \geq n$ the setting  it achieves the lower bound is simply when $n=m$ (i.e. there is exactly one item per bidder). As the number of items increases the mechanism achieves more utility in expectation. In the real world this simply translates to the intuitive fact that as you are more constrained by capacity your less able to achieve optimal welfare
>
> As for the $n>m$ case we can give some intuition for the example in our tightness result (a setting where it is hard to achieve utility close to welfare). This setting shows that a difficult case is when each bidders value distribution for their favorite item is exponential. What is difficult about an exponential distribution is that it is the middle point between MHR distributions, in which the optimal mechanism is a lottery and anti-MHR distributions, in which the optimal mechanism is a second-price auction. This leads to a case where neither of these two mechanisms do a great job at approximating social welfare.

---

### Decision · Program_Chairs · 2025-09-17

**Decision:**

Accept (spotlight)

**Comment:**

The paper studies utility maximization in a multi-agent multi-item unit-demand setting, where the valuations of all agents over all items are iid.  The authors present two mechanisms achieving tight approximation ratios.  The reviewers all find the problem and the results nice and clean, and appreciate the technical novelty and exposition.  It is noted that a follow-up paper has been accepted at STOC 2025, and that this should not be considered a weakness of the paper.  We encourage the authors to address the constructive comments by the reviewers in the final version of the paper.